# Preoperative Navigated Transcranial Magnetic Stimulation: New Insight for Brain Tumor-Related Language Mapping

**DOI:** 10.3390/jpm12101589

**Published:** 2022-09-27

**Authors:** Federica Natalizi, Federica Piras, Daniela Vecchio, Gianfranco Spalletta, Fabrizio Piras

**Affiliations:** 1Laboratory of Neurophychiatry, IRCSS Santa Lucia Fundation, Via Ardeatina 306, 00134 Rome, Italy; 2Department of Psychology, “Sapienza” University of Rome, Via dei Marsi 78, 00185 Rome, Italy; 3PhD Program in Behavioral Neuroscience, Sapienza University of Rome, 00161 Rome, Italy

**Keywords:** TMS, neuro-oncology, navigated, brain tumor, preoperative mapping, language

## Abstract

Preoperative brain mapping methods are particularly important in modern neuro-oncology when a tumor affects eloquent language areas since damage to parts of the language circuits can cause significant impairments in daily life. This narrative review examines the literature regarding preoperative and intraoperative language mapping using repetitive navigated transcranial magnetic stimulation (rnTMS) with or without direct electrical stimulation (DES) in adult patients with tumors in eloquent language areas. The literature shows that rnTMS is accurate in detecting preexisting language disorders and positive intraoperative mapping regions. In terms of the region extent and clinical outcomes, rnTMS has been shown to be accurate in identifying positive sites to guide resection, reducing surgery duration and craniotomy size and thus improving clinical outcomes. Before incorporating rnTMS into the neurosurgical workflow, the refinement of protocols and a consensus within the neuro-oncology community are required.

## 1. Introduction

The preoperative mapping of cortical and subcortical brain damage before tumor surgery is becoming an increasingly important goal. Such mapping allows for better surgical planning, more accurate risk–benefit assessments, and a lower incidence of postoperative deficits [1]. Thus, functional neuro-oncology aims to study and preserve brain functions in order to ensure a better quality of life [2]. This is especially important for tumors involving eloquent circuits, for example, the cortical/subcortical networks that control the cognitive and motor functions underlying communicative abilities, which are considered indispensable for daily life. In the past, the localizationist model assumed that cortical areas were specialized for neurological and cognitive functions [3]. Recent brain mapping methods have since been used to propose the connectionist model. According to this model, the large intra- and interindividual clinical variability and the different therapeutic responses to similar brain damage depend on the link between the cortical and subcortical areas [4]. These properties are explained by neuroplasticity, i.e., the brain’s ability to reorganize in the case of acquired damage or tissue removal [5]. Cortical structures seem to be more plastic, whereas white matter bundles seem to have less plastic potential [6]. In fact, slow-growing tumors give the brain more time to reorganize, thus delaying the onset of cognitive symptoms [7]. In contrast, fast-growing tumors cause major neurological deficits and cognitive impairments, suggesting that compensatory mechanisms take time to counteract the tumor’s effect on the cerebral circuitry [8]. These mechanisms reflect the variations in brain plasticity responses that add complexity to the onset of neuro-oncological symptoms.

Preoperative brain mapping methods, along with cognitive evaluations, allow for the assessment of functional outcome. They allow us to monitor the evolution of deficits over time and investigate the association between a loss of function and plastic adaptations [9]. Among other preoperative brain mapping methods, repetitive navigated transcranial magnetic stimulation (rnTMS) represents an important advance in the field of neuro-oncology. Indeed, it allows one to both map eloquent areas prior to surgery and guide the intraoperative direct electrical stimulation (DES) of cortico-subcortical areas [10]. The gradual introduction of TMS in neurosurgery has been the result of important scientific development. First, some evidence suggested that functional imaging did not fully meet the accuracy requirements for the preoperative planning of language mapping [11]. Second, a theoretical shift from the localizationist approach to the connectivity model occurred. As such, function is now conceptually represented as extended and dynamic networks [12]. Finally, technical improvements derived from the combination of brain stimulation techniques and neuro-navigation have made TMS more attractive. This review aims to demonstrate the expansion of rnTMS in neuro-oncological settings as a necessary tool for preoperative surgical planning. We refer only to the use of rnTMS in language mapping; thus, the neurobiology of language in terms of cortico-subcortical networks will only be briefly discussed. We also aim to describe the main preoperative methods and intraoperative DES with a focus on rnTMS preoperative language mapping in patients with tumors in eloquent brain language areas.

## 2. Materials

A literature search for articles published in the last 10 years (January 2013–June 2022) was performed using the PubMed, Google Scholar, Scopus and Web of Science databases based on the following keywords: “TMS”, “navigated TMS”, “neuro oncology”, “language”, and “preoperative mapping”. Articles were selected according to the following inclusion criteria: (i) navigated rnTMS mapping with or without DES; (ii) navigated rnTMS in combination with DTI-FT; (iii) rnTMS language mapping; (iv) language task under rnTMS stimulation; (v) patients with tumors in eloquent language areas; and (vi) patients aged 18 years or older. The exclusion criteria included: (i) articles written in languages other than English; (ii) articles with no full text available; (iii) TMS not navigated; (iv) patients with tumors in non-language-related areas; (v) patients with no other neurological or psychiatric disorders; and (vi) pediatric samples. Reviews and meta-analyses were not included; however, their reference lists were checked in order to find additional articles. No restriction was placed on sample size, tumor pathology, WHO grade, or the type of rnTMS protocol to ensure the inclusion of relevant articles. The studies were first screened by reading the title and the abstract before checking for the full text. After eligibility assessment, a data extraction sheet of the selected articles was developed in order to extract sample characteristic and applied protocol. The application of inclusion and exclusion led to the selection of 20 studies.

## 3. The Neurobiology of Language: A Brief Introduction to Cortical and Subcortical Networks

The neurobiology of language has a long history that begins with studies conducted in the 19th century by Paul Broca. Later, Wernicke’s doctoral thesis, Lichtheim’s diagram, and Geschwind’s reconceptualization of the language mechanism allowed for significant contributions to the era of language [13]. The Broca–Wernicke–Geschwind model provided substantial innovations in the conceptualization of language and included, in addition to the classical Broca and Wernicke areas, the involvement of the white matter bundles connecting these two cortical regions [14]. The 20th century represented a step forward in the study of language neural architecture owing to the development of structural imaging technologies, including computed tomography and magnetic resonance imaging. These advances made it possible to study pathology in vivo and contributed substantially to the birth of the neurobiology of language, resulting in a confluence of different disciplines. Although Broca’s studies marked the history of classical aphasiology, with the introduction of these new techniques, it has been possible to demonstrate that the areas involved in language processing extend to a large part of the subcortical bundles [15]. For instance, Crosson (2021) showed that the cortico-thalamo-cortical and cortico-cortical circuits support certain language functions such as auditory-verbal comprehension and word retrieval [16]. In particular, the left thalamus seems to play a role in the manipulation of lexical information, as demonstrated by cases of aphasic patients with thalamic damage (for a review, see [17]). Other studies have focused on the arcuate fasciculus (AF), as it represents an important white matter association pathway that connects Broca’s and Wernicke’s areas bilaterally [14] (Figure 1). Subsequently, advances in neuroscience and neuroanatomy have allowed us to surpass the idea of AF as the only link connecting these two areas [18]. For instance, the dual stream model suggests that some language abilities may depend more on the functional interaction between cortical regions than on their functional specializations [19]. This model does not comprehensively explain the spectrum of aphasic disorder; however, Hickok and Poeppel’s work helped form a new theoretical framework for a language connectome in which language functions would emerge from the dynamics of the connections between cortical and subcortical areas that may not have properties specific to language itself [18]. Within this theoretical framework, the importance of AF has been consistently demonstrated by intraoperative brain mapping and lesion studies (e.g., [20]). A recent study in tumor patients demonstrated that the onset of postoperative aphasia was associated with a distance from the resection border to the AF of less than 5 mm [21]. Another study reported the preservation of language skills in patients with AF integrity, as opposed to patients with a loss of AF fibers who manifested non-fluent aphasia [10]. These studies demonstrate that damage to the AF white matter bundles, and in general to subcortical traits, may be crucial to the onset of aphasic disorders. Therefore, the study of cortico-subcortical language networks represents a major challenge in understanding tumor-induced compensatory mechanisms.

## 4. Preoperative Brain Mapping Methods

The last decade has been revolutionary from the perspective of brain tumor neurosurgery, and advances in clinical neuropsychology have allowed for the establishment of preoperative brain mapping protocols.

Among these preoperative methods, functional magnetic resonance imaging (fMRI) is the most widely used in mapping brain function [22]; in particular, task-based fMRI is an increasingly utilized protocol in clinical trials with neuro-oncological patients. Magnetoencephalography (MEG) represents another non-invasive approach that aids in surgical planning and the identification of language lateralization [23]. However, the development of MEG applications is still limited due to the high apparative costs and the high level of staff expertise required [24]. Diffusion-tensor imaging tractography (DTI-T) allows for the detection and characterization of white matter bundles to explore brain connectivity [25]; in addition, in neuro-oncological patients, it is widely used to define white matter tracts using DTI fiber tracking (DTI-FT). Preoperative or intraoperative DTI-FT protocols have been compared with subcortical stimulation mapping and have shown excellent intraoperative correlation with direct subcortical stimulation [26].

Compared to these preoperative techniques, TMS is emerging as a method that can provide valuable data. TMS is delivered using a coil, typically with eight configurations, to generate a magnetic field in a specific area of interest.

The magnetic trains of pulses temporarily interrupt the brain’s function, causing a transient disruption in the patterns of neural activity [1], or a “virtual lesion” [27]. While the mapping of motor functions uses single pulses, that of language abilities uses short bursts of TMS pulses (repetitive TMS or rTMS) which, when guided by neuro-navigation technology, make navigated stimulation possible (rnTMS). The neuro-navigation system generally consists of a device that keeps track of the patient’s brain locations, surgical instruments, a console and display where images and other navigation information are projected, and, finally, other navigation accessories [28]. Specifically, neuro-navigation is an image guide that aids the surgeon in both pre-surgical planning and surgery by using pre-acquired MRI data to locate the intracranial pathology (Figure 2).

rnTMS has many advantages, including the ability to be carried out multiple times in a controlled environment, unlike intraoperative DES (discussed below), in which fatigue during the awake craniotomy represents a strong limiting factor. In addition, when mapped intraoperatively, the cortical surface is limited to the degree of cerebral cortex exposure from the craniotomy [29], an issue that does not arise in rnTMS. However, DES is still thought to be the gold standard among most neurosurgeons since the direct stimulation of exposed brain regions is considered the best method to map compromised areas.

## 5. Intraoperative Direct Electrical Stimulation

One of the most important challenges during neurosurgery is the localization of tumor tissue with high spatial accuracy in order to remove it while minimizing its impact on cognitive functions [30]. Unfortunately, the ability of surgeons to perform total resections has been often compromised by traditional operational methods [30]. Studies have reported that the prevalence of permanent damage to neural function (no recovery within 1 year after operation) is 13–27% when conventional surgical methods are used [30]. To improve clinical outcomes, the advanced techniques of neuro-navigation and awake craniotomy have been used to map neural function.

Direct electrical stimulation (DES) is performed during awake craniotomy, i.e., an anesthesia technique that allows patients to regain consciousness during surgery; it consists of the electrode stimulation of exposed fiber bundles and cortical portions while performing a cognitive task during surgical intervention [31]. The “2 out of 3” rule is generally used to determine whether an area is eloquent or non-eloquent for language (or any other function), and it implies that a cortical area is stimulated three times during the execution of a task. If a performance error occurs at least twice during DES stimulation, then that area is considered as eloquent for the function tested [32]. In order to better define eloquent and non-eloquent areas, a recent study proposed a three-level system based on the cortical, subcortical, and clinical features of language eloquence to allow for a more accurate comparison of language tumors (for more details, see [33]).

Many studies have confirmed the accuracy of DES and compared it with other noninvasive methods. In particular, comparisons between fMRI and DES have yielded conflicting results, partly due to the methodological differences in these two methods. fMRI is based on the statistical analysis of regional changes in oxygenated blood, which is impaired in the presence of intracerebral lesions; this hampers the accuracy of fMRI [11]. In light of these findings, DES and rnTMS have already replaced fMRI in some institutions [34]. However, DES, in addition to being invasive, has other disadvantages, such as patient noncompliance. In these cases, for a successful intraoperative mapping, managing patient expectations is vital [35]. Despite its limitations and invasiveness, to date, DES remains the gold standard, since it allows for the definition of the functional boundaries of resection in glioma surgery [36]. However, for those neuro-oncological patients who are not candidates for awake craniotomy, in order to guide surgical resection, it is important to identify a standardized and safe preoperative mapping protocol [37], especially for those with language tumors.

## 6. rnTMS Language Mapping: A Promising Approach in Language Neuro-Oncology

Pascual-Leone and colleagues (1991) were among the earliest to demonstrate the ability of TMS in inducing speech arrest in epileptic patients; the same lateralization results were also obtained with the WADA test (intracarotid amobarbital test), which is the gold standard for investigating language dominance in epileptic patients [38]. However, subsequent studies demonstrated inconsistencies between TMS and WADA data (e.g., [39]). This raised the need to refine TMS and provided fertile ground for the development of neuro-navigation systems. Such an effort was aimed at enabling a more detailed investigations of preoperative language mapping and a more accurate prediction of hemispheric dominance [40].

Early studies with TMS focused on motor mapping and only few mapped language areas, since they involve a complex network that is difficult to localize and, therefore, to map [41]. However, preoperative language mapping represents an area of interest and clinical utility. The results of these studies with respect to the use of different language tasks and applications in particular clinical contexts will be discussed below.

### 6.1. Object vs. Action-Naming Protocol

Object naming is the most commonly used language task during rnTMS stimulation; it requires the patient to name objects presented as drawings or pictures on a monitor placed in front of him/her. Initially, the object-naming task is performed without rnTMS stimulation (baseline recording) and all incorrectly named objects are discarded from the subsequent sequence; this establishes the patient’s familiarity with the words that will be produced during the following rnTMS mapping session (Lefaucheur and Picht 2016; Krieg and others 2017). After baseline, object naming is performed under rnTMS stimulation, and the session is recorded to assess the subject’s performance and language errors (Corina and others 2010). Finally, language errors are assigned to their respective anatomical locations [42,43]. The rnTMS protocol generally involves a train of pulses of 5 to 10 Hz administered from 0 to 300 msec after the presentation of each object. One group of authors showed that an onset protocol (pulse starting at 0 msec after object presentation) generated lower prediction errors, i.e., false positives and false negatives, in the cortical response patterns than the delayed protocol (pulse starting at 300 msec after object presentation) [44].

By using only object-naming tasks, more complex abilities that help to delineate the neuroanatomy of language could be overlooked [45], since object naming does not engage other processes that are important for conveying linguistic information. For example, verbs are relevant to everyday communication since they include information about the argument structure and thematic roles [46]. Consequently, some studies have proposed the inclusion of an action-naming task [47] to investigate grammatical and semantic abilities, which are known to rely on anatomically distinct circuits [48]. This task requires the patient to name events using an infinitive, gerund, or inflected verb form [49]. Ohlerth and colleagues compared the sensitivity of object vs. action-naming tasks for error elicitation in different cortical surfaces under rnTMS stimulation in 20 healthy participants. The results showed that action naming, under repetitive TMS stimulation, provided a higher error rate than object naming in both hemispheres. The retrieval of verbs is more easily perturbed by rnTMS than the retrieval of a noun [50], suggesting, in line with DES data, that object naming may be insufficient for accurately mapping language circuits [51]. The authors also provided an error comparison that identified the difference in error rates during action vs. object naming and showed that errors occurred more frequently during the naming of actions than objects. Therefore, rnTMS seems to affect verbs more than nouns, probably because the conceptual and lexico-semantic information of verbs is more widely distributed at the cortical level [50]. Another study showed that rnTMS induced a larger number of errors with transitive compared to intransitive verb in the left hemisphere, in twenty healthy people. This may reflect the fact that the production of finite transitive verbs requires more complex lexico-semantic syntactic processes and a larger amount of argumentative structural information than intransitive verbs [49].

Very few studies have used rnTMS with both object and action-naming tasks in language tumor patients to investigate the efficacy of this protocol in accurately mapping eloquent regions as to prevent postoperative language deficit. A recent experiment featuring a dual naming task was conducted by Ohlerth and colleagues (2022), who incorporated the rnTMS protocol into the pre, intra, and postoperative workflow in seven patients with tumors in eloquent language areas. The study suggested that the action-naming task was more sensitive in detecting minor pre-existing language impairments preoperatively and in predicting intraoperative positive mapping regions [45].

Most studies use a qualitative analysis of speech errors, generally categorized according to the scheme proposed by Corina [52]. In TMS motor cortex mappings, the evoked responses can be quantitatively monitored by electromyography recordings; however, no such setup exists for language mapping. In this regard, the use of an accelerometer to detect vocalization-related larynx vibrations combined with an automatic routine for voice onset detection can provide quantitative additional data [53]. This approach was found to be feasible, since it detects the rnTMS stimulus train onset, the corresponding vocalization onset, and non-response errors [53].

A recent avenue of research has featured attempts to perform a quantitative assessment using reaction time and picture–word interference (PWI) paradigms. In particular, PWI requires the participant to name the target while a visual or auditory distractor word is presented. A recent study adapted the PWI naming paradigm for rnTMS language mapping in 30 healthy subjects and showed that the presentation of a visual distractor induced higher facilitation effects than an auditory distractor [54]. In this sense, participants revealed a preference for the unimodal presentation mode in which the visually written distractor word appeared simultaneously with the target picture presentation [54]. Both approaches to data analysis—quantitative naming latencies and qualitative naming errors—may aid in the future assessment of language, especially for neuro-oncological language patients.

### 6.2. Further Applications of rnTMS

Detailed preoperative language mapping is also important in the case of bilingual patients, a peculiar subgroup that may be characterized by a different cortical-subcortical representation of languages. Baro and colleagues reported a patient with a tumor in the left temporal lobe who, having refused awake craniotomy, underwent preoperative rnTMS-based DTI-FT language mapping. As previously explained, DTI-FT is used to study the subcortical white matter and single traits relevant for language function. The results showed convergence in the posterior areas of the first and second language pathways, an overlap that was interpreted in light of the patient’s high second language proficiency, as demonstrated by a neuropsychological assessment [55].

Modern neuroscience has increasingly enabled the identification of areas involved in language; for example, a previous study suggested that the posterior middle frontal gyrus (MFG) is an important center of cortical integration for the dorsal and ventral flows of language [56]. More importantly, it was demonstrated a cluster of positive responses (language errors elicited by DES) in the MFG to both rnTMS and DES stimulation in tumor patients [56].

The previous studies mentioned refer to cortical tumors. In recent decades, the subcortical representation of language has received increasing attention. In fact, cases with damage to subcortical traits show more severe language loss than those with cortical damage [57]. However, rnTMS is not suitable for investigating deep subcortical structures because of its physical limitations [42].

Recent innovations have enabled subcortical mapping through the combination of rnTMS and DTI-FT. For this type of tracking, information in the regions of interest (ROIs) is needed to identify start and/or end points at the cortical level that are connected by the subcortical tracts. Specifically, the rnTMS stimulation of cortical areas allows for positive/eloquent areas to be identified and defined as ROIs in order to visualize subcortical tracts with subsequent tractography [10]. The combination of rnTMS with DTI-FT has been successfully refined in recent years, leading to remarkable results from the clinical perspective [58]. Indeed, neuro-oncology patients undergoing rnTMS-based DTI-FT showed fewer deficits at discharge when compared to a control group [59].

A study using nrTMS and DTI-FT raised the question of whether it is possible to achieve a better visualization of fiber tracts by adding the action-naming task [60]. The authors showed that the action-naming task with rnTMS mapping and subsequent tracking led to a better visualization of the language subcortical network in healthy volunteers. Moreover, mapping and tractography with object and action naming is particularly useful for studying brain reorganization induced by tumor growth [60]. Another study evaluated the reliability of rnTMS in mapping the functional architecture of the AF in patients with tumors in the perisylvian regions. The authors showed that language errors occurred mainly when stimulating the frontal and parietal regions and corresponded in all cases to at least one anatomical termination of the AF. Their findings suggested that the rnTMS-based DTI-FT protocol could be useful for investigating the subcortical architecture of language-related areas in cancer patients [61].

What remains to be understood is which DTI-FT approach is most efficient from a clinical perspective. In this regard, a recent study performed DTI-FT on major lingual pathways using five distinct approaches: (Ia) based on anatomical landmarks; (Ib) based on lesion-focused landmarks; (IIa) based on rTMS; (IIb) based on rTMS with post-processing; and (III) enhanced with rTMS (based on a combination of structural and functional data). The results showed that the post-processing of rTMS-based tractograms (IIb) did not improve its utility for surgical planning and risk assessment. In contrast, high accuracy indices were observed for the anatomy-based approaches (Ia and Ib), as they were most successful in delineating key linguistic features. Therefore, lesion-focused landmark-based approaches (Ib) and enhanced rTMS (III) were suggested as the preferred methods [62].

Newer approaches include the combination of graph theory, rnTMS, and the connectome analysis of DTI data. This is a multidisciplinary paradigm that considers the brain as a complex network and allows for the study of the local and global effects of gliomas on complex networks [63]. Indeed, previous studies using rnTMS-based FT have focused only on single neural tracts in patients with glioma-induced aphasia [64]. However, recent studies have shown that gliomas have a global impact on the entire brain [65]. In this regard, a recent study aimed to evaluate the global and local properties of function-specific connectomes derived from linguistic mapping with rnTMS in patients with or without aphasia. The results showed that the global interference of gliomas in the distribution and performance of various brain networks leads to the clinical manifestation of different levels of aphasia [66]. Alterations in the efficiency of various networks between patients with and without aphasia reflect the interaction of different brain regions in the transmission of information, supporting the theory of language function based on the link between dorsal and ventral streams [67].

Several factors may induce different responses to rnTMS stimulation, such as the presence of pathology and cortical reorganization following damage. Language functions are traditionally associated with the left hemisphere, which is defined as dominant. However, several recent studies have demonstrated the importance of the right hemisphere in language tasks, such as in patients with slow-onset pathologies in the left hemisphere [68]. Indeed, a brain lesion in the dominant hemisphere can induce the cortical reorganization of the network involved in language processing, with the activation of homologous cortical areas and the functional reorganization of perilesional areas. In this regard, a study mapped language areas with rnTMS in subjects with tumors in language areas vs. healthy subjects to assess the differences in language representation. The results showed that the rnTMS-induced naming error rates in the right and left hemisphere were quite similar [69], not supporting left hemisphere lateralization [70]. Therefore, the right hemisphere may have a plastic ability to take over language functions in the presence of a left hemisphere lesion.

Another study investigated whether other factors, such as age, gender, the histology of the lesion, cognitive impairment, and aphasia, could influence the intrinsic random error rate during rnTMS language mapping. Schwarzer and colleagues (2018) showed that the main influencing factors were cognitive impairment and aphasia, as these patients made more errors during baseline and stimulation naming. This emphasizes the importance of distinguishing naming errors under stimulation from those caused by other dysfunctions. This study confirmed the suitability of most patients for preoperative rnTMS language mapping, but also underlined that the baseline naming should always be implemented and should be further optimized, especially for severe aphasia and cognitive impairment patients [71].

These studies demonstrated the increasing expansion of preoperative rnTMS in neurosurgery and, for this reason, it is critically important to ascertain that the procedure is safe and well tolerated. In this regard, a systematic assessment of the risks associated with rnTMS in neuro-oncological patients showed minimal risk of pre-surgical rnTMS language mapping and assured its safety and tolerability [72]. However, some studies show differences in rnTMS language mapping in terms of the cortical distribution of language errors, eloquence sites detected, comparison with DES, or postoperative outcome. These differing results might depend on the pathological and tumor characteristics of the patients (see Table 1) and/or the language rnTMS mapping protocol (see Table 2), suggesting that language rnTMS accuracy mapping may vary.

## 7. Discussion

Language is a uniquely human capacity that should be preserved in tumor surgery. The resection of tumors located in eloquent language areas, however, carries a high risk of surgery-related language impairment. Since not all people are candidates for awake craniotomy, several disciplines must cooperate to identify a preoperative mapping method that can guide tumor resection and limit postoperative language deficits. Since its earliest applications in the motor cortex, TMS has gained considerable success in clinical settings. The introduction of navigation systems has made it possible to visualize brain areas with better accuracy. Thus, the evidence supporting this method is now outpacing that of fMRI mapping data [11].

The literature reveals that rnTMS based on action naming, as opposed to object naming, is more accurate in detecting preexisting speech disorders and positive intraoperative mapping regions [45]. In terms of region extent (EOR) and clinical outcome, rnTMS shows accuracy in identifying positive sites to guide resection [33], reducing surgery duration and craniotomy size and thus improving clinical outcomes [78]. Compared with DES, rnTMS shows high sensitivity, specificity, and positive predictive value in identifying the eloquent language areas in patients with tumors [32,73,76]; when compared with fMRI, rnTMS is a more sensitive but less specific tool for preoperative language mapping than DCS, especially when using the “2 out of 3” rule and a PTI of 0 msec [34]. rnTMS also provides insight into functional reorganization in the face of neuroplasticity, as demonstrated by its high correlation with DES data [75].

Since white matter damage can lead to a permanent language deficit [10], the addition of DTI-FT has been proposed to investigate the subcortical features of language, such as the AF [61], and the language pathway in bilingual patients [55]. In particular, the rnTMS-based DTI-FT protocol can reveal information on individual risk assessment for surgery-related decline in language function [74]. This protocol also allows for the reconstruction of damages in the cortico-subcortical language network in those patients with tumors in subcortical language tracts who cannot undergo awake craniotomy [77]. Several studies have shown that the rnTMS-based DTI-FT protocol is accurate in predicting postoperative deficit and shows high sensitivity and specificity [59,77]. These promising data make brain tumor resection guided only by rnTMS-based DTI-FT mapping results feasible [37]. Therefore, when intraoperative DES is not possible, rnTMS-based DTI-FT may guide a surgery strategy with an optimal functional outcome, as demonstrated by post and follow-up outcomes [10].

rnTMS has been shown to be useful for all patients; however, those with preexisting aphasia or cognitive impairment commit significantly more language errors, despite baseline stratification [71]. This opens future issues about the possibility of considering a different rnTMS language protocol for these patient groups. Regarding safety and tolerability, rnTMS has been shown to be well tolerated in neuro-oncological patients [72], although there is a need to expand these studies to patients with high seizure frequency.

Despite the promising results of rnTMS, these studies underscore the need for protocol refinement and a consensus regarding neuroplasticity, behavioral measurement, and qualitative/quantitative analysis in brain tumor patients. Gaps in our knowledge regarding neuroplasticity make it difficult to understand cortico-subcortical reorganization following acquired brain injury and the best methods to map compromised areas. A recent comparison with DES data showed variability in the sensitivity and specificity of rnTMS language mapping results [29]. Specifically, the different values appeared to depend on the criteria that were used to determine whether a cortical area was considered language eloquent or non-eloquent, for which advances have been already proposed [33]. Although language glioma resection based solely on rnTMS data has been accomplished [33], there is no final evidence that rnTMS can replace DES. In particular, it should be used in combination with other preoperative methods to aid surgical planning.

Taken together, these results suggest that in the near future, language tumor resection could be driven solely by rnTMS language mapping. This will avoid high costs, invasiveness, and the ineligibility of patients to intraoperative DES. This requires further studies and a consensus in the neuro-oncology community about the best protocol to map eloquent language areas compromised by tumors.

## 8. Limits

Considering that this was not a systematic review, it has inherent limitations.

Although every effort has been made and selection criteria have been defined, it is possible that some studies may not have been included in the present review, for example, non-English articles and studies that have been published in non-article form (e.g., conference papers). Our aim was to provide a narrative review focused on navigated rnTMS language mapping applied only to patients with tumors in language-related areas. Our purpose, together with the selection criteria, restricted the eligibility of studies. Nevertheless, we thought it would be useful to contribute to a general overview of such a specific issue of rnTMS development over the past decade. Future studies could use this review as reference point about the current language protocols to investigate language subcomponents.

Moreover, the heterogeneity of findings among the cited studies could be related to the different methodologies employed. Indeed, the studies used different sample selection, recruitment criteria, and outcome measures. In particular, the age range varied, and the heterogeneity of pre, intra, and post language assessments results may have depended on concomitant adjuvant therapy, tumor grade, the extent of infiltration, and language reorganization. Such variables should be carefully taken into account in order to accurately estimate the postoperative sparing of language skills. Resolving these issues requires consensus in the awake surgery/neuro-oncology community, for which some attempts have been made in Europe [79].

## 9. Conclusions and Future Directions

The shift from the localizationist model to dynamic and interconnected network models has allowed for a multidisciplinary approach to the treatment of neuro-oncological patients. Future studies can deepen the complexity of the brain network by adding tasks that also evaluate linguistic subcomponents. Further studies will also be necessary to define the parameters of stimulation and the refinement and implementation of tasks sensitive to rnTMS. Longitudinal studies with larger samples are needed to understand the link between functional changes and the progression of tumors. Given the large intraindividual variability and neuroplastic mechanism, all tumors presumed to be located in eloquent language areas should be routinely mapped through the combination of rnTMS and other preoperative methods.

## Abbreviation

TMStranscranial magnetic stimulationrTMSrepetitive transcranial magnetic stimulationrnTMSrepetitive navigated transcranial magnetic stimulationDESdirect electrical stimulationfMRIfunctional magnetic resonance imagingrs-fMRIresting state functional magnetic resonance imagingDTI-Tdiffusor tensor imaging tractographyDTI-FTdiffusor tensor imaging fiber trackingMEGmagnetoencephalographyWADAintracarotid amobarbital testRMTresting motor thresholdROIsregion of interestEORextent of resectionAFarcuate fasciculus

## Figures and Tables

**Figure 1 jpm-12-01589-f001:**
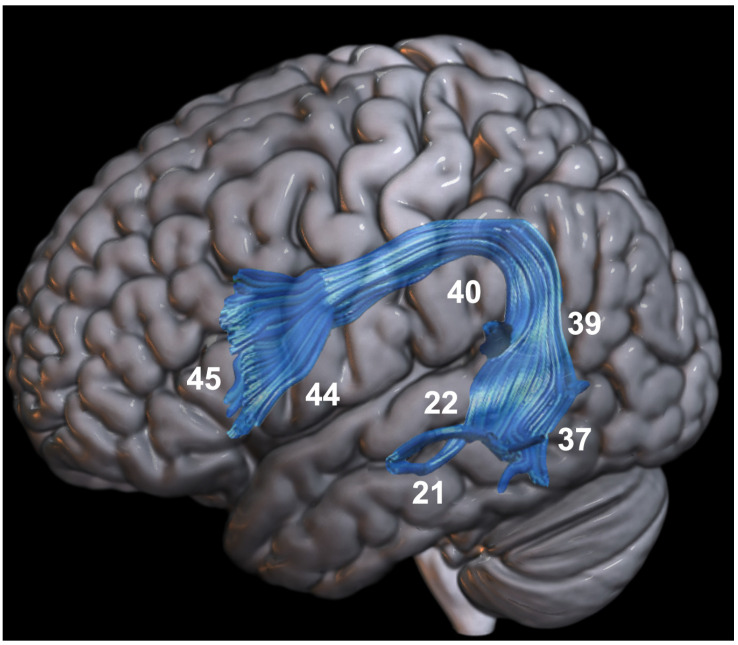
The arcuate fasciculus (blue) connecting the anterior and posterior language areas. White numbers refer to Brodmann’s areas.

**Figure 2 jpm-12-01589-f002:**
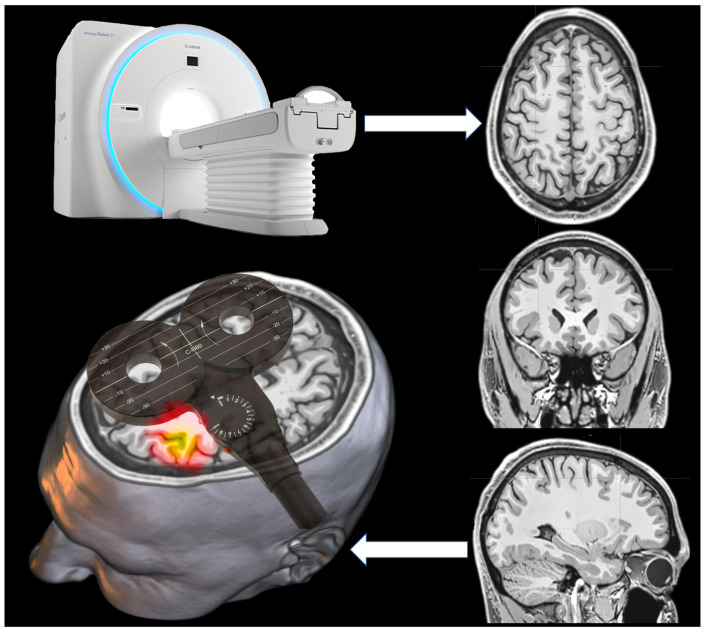
TMS through neuro-navigation.

**Table 1 jpm-12-01589-t001:** Sample characteristics of the selected studies.

[Study], (N)	rnTMS	DES	Patient(s) Characteristic		Tumor Characteristics		
			Gender	Age Range	WHO Grade *	Tumor Location	Pathology
[45], (7)	Yes	Yes	6M; 1F	33–70	II/III	SFG, prG, post STG, parietal white matter, mid ITG, insula	A AA OD GBM M of cervix carcinoma
[55], (1)	Yes (+ rnTMS-based DTI-FT)	No	F	54	IV	AG	n/a
[56], (24)	Yes	Yes	12M; 12F	18–74	II/III/IV	Left frontal	AA AOD DA GBM GNT OD HHG LGG
[33], (147)	Yes	Yes	79M; 68F	20–84	I/II/III/IV	Frontal, insular, temporal, parietal	n/a
[66], (60)	Yes (+DTI)	n/a	17M; 43 F	43–73 (Group 1); 52–76 (Group 2)	I/III/IV	Left perisylvian region adjacent to AF	
[61], (28)	Yes (+DTI)	n/a	14M; 14F	28–77	I/II/III/IV	Frontal, temporo-parieto-occipital, temporal	GG OA OD A Glioma M Ependymoma
[73], (61)	Yes	Yes	39M; 22F	18–72	II/II/III/IV	Left/right frontal, temporal, parietal, insular	GBM AO AA OG DA Pleomorphic xanthostrocytoma Dysembroplastic neuroepithelial tumor Pilocytic astrocytoma
[74], (98)	Yes (+ rnTMS-based DTI-FT)	n/a	51M; 49F	19–83	I/II/III/IV	Not reported	Arteriovenous Malformation M CAV
[75], (18)	Yes	n/a	Not reported	18–75	II/III/IV	Left side ant/post perisylvian areas	GBM AA OD DA
[76], (11)	Yes	Yes	5M; 6F	29–65	II/III/IV	Frontal, parietal, temporal, insula	HHG LGG
[10], (10)	Yes (+ rnTMS-based DTI-FT)	Yes	Not reported	31–74	II/III/IV	ant SMG, insular, mid MFG, post MFG, op IFG, STG	GBM AA DA
[77], (20)	Yes (+ rnTMS-based DTI-FT)	No	14M; 6F	38–77	II	Left fronto opercular, left fronto insular, left temporo insular, left temporo parietal, left parietal, left temporal, left frontal	OD DA GBM M
[71], (101)	Yes	n/a	56M; 45F	21–81	II/III/IV	Left/right temporal, parietal, frontal, insular	LGG HHG Meningioma Vascular Malformations
[59], (7)	Yes (+ rnTMS-based DTI-FT)	Yes **	6M; 1F	39–76	n/a	Left fronto- insular, left frontal, left temporal, right/left fronto- opercular	DA OA OD
[37], (4)	Yes (+ rnTMS-based DTI-FT)	n/a	2M; 2F	27–52	III/IV	Mid STG, op IFG, post MTG, prG	OD GBM A CAV
[72], (196)	Yes	n/a	89M; 107F	15–79	II/III/IV	Frontal, parietal, temporal, temporo-mesial-insular	CAV AVM M Meningioma
[34], (35)	Yes (+ fMRI)	Yes	22M; 13F	24–74	I/II/III/IV	Op IFG, AG, tr FG, mid MTG, post MTG, ven prG, post STG, mid MFG, ant SA, mid STG, ant STG, mid prG	CAV AA GBM DA AVM GNT OA
[78], (50)	Yes	Yes	33M; 17F	36–58 (Group1); 30–60 (Group2)	I/II/III/IV	Posterior/anterior language related areas	AVM Metastasis
[69], 50	Yes	n/a	31M; 19F	20–75	I/II/III/IV	Left temporal, parietal, frontal	n/a
[32], (20)	Yes	Yes	9M; 11F	36–60	I/II/III/IV	Left frontal, left temporal, AG	A GBM CAV

Study (N) and patient(s) and tumor characteristics of each included study using rnTMS with or without DES, DTI-FT and fMRI for language mapping in patients with tumor in eloquent language areas. Abbreviations: WHO—World Health Organization tumor grade. Tumor location: post—posterior; ant—anterior; mid—middle; tr—triangular; op—opercular; ven—ventral; SFG—superior frontal gyrus; STG—superior temporal gyrus; SMG—supramarginal gyrus; ITG—inferior temporal gyrus; IFG—inferior frontal gyrus; MTG—middle temporal gyrus; MFG—middle frontal gyrus; prG—precentral gyrus; postG—postcentral gyrus; SG—supramarginal gyrus; IF—inferior frontal; FG—frontal gyrus; AG—angular gyrus; IC—insular cortex. Pathology: HHG—high grade glioma; LGG—low grade glioma; M—metastasis; OD—oligodendroglioma; OA—oligoastrocytoma; AA—anaplastic astrocytoma; GBM—glioblastoma; A—astrocytoma; DA—diffuse astrocytoma; AOD—anaplastic oligodendroglioma; AOA—anaplastic oligoastrocytoma; AGG—anaplastic ganglioglioma; GG—ganglioglioma; GC—gangliocytoma; PA—pilocytic astrocytoma; CAV—cavernoma; AE—anaplastic ependymoma; L—lymphoma; AVM—arteriovenous malformation; GNT—glioneuronal tumor. n/a—not available. * Information on the tumor entity based on histopathological evaluation and WHO grading. ** DES applied only in the case of tumors located in motor areas.

**Table 2 jpm-12-01589-t002:** rnTMS protocols used in the selected studies.

[Study]	rnTMS Protocol						
	Frequency (Hz)	Pulse Train	Display Time and Inter-Picture Interval (ms)	PTI (ms)	Device	Task (Number of Items)	Rule
[45]	5	5	700 (ON)/1000 (AN) and 2500	0	Nexstim eXimia NBS system (5.1 version)	ON (80) and AN (75)	2 out of 3
[55]	5/10	5/10	700 and 2500	0	Nexstim Oy NBS system (4.3 version)	ON (80)	2 out of 3
[56]	n/a	n/a	500–1000 and 2500	n/a	n/a	ON and AN (n/a)	2 out of 3
[33]	n/a	n/a	n/a	n/a	n/a	ON (n/a)	n/a
[66]	5	5	n/a	n/a	Nexstim eXimia NBS system (5.1.1 version)	ON	n/a
[61]	5/7	5/7	1000 and 2500–4000	0	Nexstim NBS system (4.3 version)	ON (150)	1 out of 3
[73]	5/7/10	10	700 and 2500	300	Magstim Rapid	ON (40)	2 out of 3
[74]	5	5	n/a	n/a	Nexstim NBS system (4.3 or 5.0 version)	ON (n/a)	n/a
[75]	5/7/10	5/7/10	700 and 2500	n/a	eXimia NBS system (4.3 version) and NEXSPEECH module	ON (n/a)	n/a
[76]	5	5	700 and 2500-3000	0	eXimia NBS and NEXSPEECH module	ON (n/a)	2 out of 3
[10]	n/a	n/a	n/a	n/a	eXimia NBS (4.3 version) and NEXPEECH module	ON (n/a)	n/a
[77]	5/7/10	5/7/10	700 and 2500	n/a	Nexstim NBS system (4.3 version)	ON (n/a)	2 out of 3
[71]	5/10	n/a	700-1000 and 2500-4000	n/a	eXimia NBS system (3.2.2 version) and Nexstim NBS (4.3 version)	ON (150)	n/a
[59]	5	5	4000 and 4000	0	NexstimOy NBS system (4.3 version)	ON (n/a)	2 out of 3
[37]	5	5	n/a	0	NexstimOy NBS system (4.3 system) and NEXSPEECH module	ON (131)	n/a
[72]	5/7/10	5/7/10	1550-3000 and 3000-5000	n/a	Nexstim NBS system	ON (120)	n/a
[34]	5/7	5/7	700 and 2500	0/300	eXimia (3.2.2 version) and Nexstim NBS (4.3 version) and NEXSPEECH module	ON (131)	2 out of 3
[78]	5/7	5/7	700 and 2500	n/a	eXimia (3.2 version) and NEXSPEECH module	ON (131)	n/a
[69]	5/7/10	5/7/10	n/a and 2500	300	eXimia (4.3 version) and NEXSPEECH	ON (150)	n/a
[32]	5/7/10	5/7	n/a and 2500	300	eXimia (3.2.2 version) and Nexstim NBS (4.3 version) with a NEXSPEECH module	ON (122)	2 out of 3

rnTMS protocol (frequency and pulse train, display time and inter-picture interval, PTI, device, task, and eloquence rule) of studies using rnTMS language mapping in patients with tumors in eloquent language areas. Abbreviations: n/a—not available; ON—object naming; AN—action naming; PTI—picture trigger interval.

## Data Availability

Not applicable.

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
