# Peer review of "Preoperative Navigated Transcranial Magnetic Stimulation: New Insight for Brain Tumor-Related Language Mapping"

_jpm, 2022, doi:10.3390/jpm12101589_

Round 1

Reviewer 1 Report

The review is to examine literatures regarding preoperative and intraoperative language mapping using navigated TMS with or without direct electrical stimulation (DES) in adult patients with tumors in eloquent language areas

The authors did the electronic searches in PubMed and Google Scholar database only. The authors should also search in another 2 bigger databases – Scopus and Web of Science. The Web of Science database, for instance, included the WoS Core collection, Current Contents Connect, Derwent Innovations Index, KCI Korean Journal Database, Medline, Russian Science Citation Index, and SciELO Citation Index which covers wider area. The outcome might be different.

Line 10 – In Abstract, moder neuro oncology – modern (typo).

Line 12 – In Abstract, examines literature – literatures (plural).

Line 14 – Literature reveals that – Literatures (many literatures right?)

Line 26 – whith the aim – with the aim. Sentences must be simplified.

Line 30 to 44 - Paragraphs are too long. Split the paragraphs and rewrite.

Line 47 to 65 - Sentence connectivity is lost. No flow in reading. The writing must be in flow. Need to split the paragraphs and rewrite.

Line 76 to 83 - Provide the introduction first which must be from standard references. Then reorganize the whole paragraph.

Line 111 – Figure 1: Provide reference unless it is own diagram.

Line 118 – the most widely used “method”? Hanging sentence there…

Line 131 – Delete the reference [1] as it is also cited in line 133.

Line 143 - Figure 2: Provide reference unless it is own diagram.

Line 163 to 187 – The sentences must be continuous for easier understanding. Rewrite and organize.

Line 282 to 307 – The sentences must be written with flow in line with previous text. Separate into 2 or 3 paragraphs.

Line 360 – Literature reveals that - The noun literature can be countable or uncountable. In more general, commonly used, contexts, the plural form will also be literature. However, in more specific contexts, the plural form can also be literatures e.g. in reference to various types of literatures or a collection of literatures.

Line 360 to 388 - Split the paragraphs for better understanding.

Line 401 to 405 – Rewrite the sentences.

Line 407 to 416 – Reorganize the sentences to improve the clarity.

Line 418 to 424 – Write in simple sentences.

Table 1 and Table 2 – The literatures were arranged from latest year 2022 to year 2013. However, the fourth row is [34] from year 2015. Any reason for that?

How did the authors do the search? First search in Google Scholar (custom range of 2013-2022) using all keywords gave return of 2770 results. Did the authors read all 2770 papers or just the first 3 pages?

For PubMed, search was conducted using ((((TMS) AND (navigated TMS)) AND (neuro oncology)) AND (language)) AND (preoperative mapping) keywords and gave only one result. Alternatively, another search was conducted using ((((TMS) OR (navigated TMS)) OR (neuro oncology)) OR (language)) OR (preoperative mapping) keywords and gave 435,386 results. How did the authors conduct the search?

Overall - Write in simple sentences and check spellings with grammatical errors in the whole manuscript.

Author Response

Reviewer 1

Q1: The authors did the electronic searches in PubMed and Google Scholar database only. The authors should also search in another 2 bigger databases – Scopus and Web of Science. The Web of Science database, for instance, included the WoS Core collection, Current Contents Connect, Derwent Innovations Index, KCI Korean Journal Database, Medline, Russian Science Citation Index, and SciELO Citation Index which covers wider area. The outcome might be different.

R1: We thank the reviewer for the suggestion. We searched both databases and included additional studies. From Scopus, we included:

- Zhang et al 2021 “Elucidating the structural–functional connectome of language in glioma-induced aphasia using nTMS and DTI” in Scopus. We decide to included it since it provide new approach about graph theory model and rnTMS (reference [66], also added in table).

- Silva et a.l, 2022 “Distinct approaches to language pathway tractography: comparison of anatomy-based, repetitive navigated transcranial magnetic stimulation (rTMS)–based, and rTMS-enhanced diffusion tensor imaging–fiber tracking” to underline which is the best rnTMS DTI-FT approach to linguistic pathway (reference [62]).

From Web of Science, we include:

  • Rosler et al., 2014: “Language mapping in healthy volunteers and brain tumor patients with a novel navigated TMS system: Evidence of tumor-induced plasticity” (reference [69], also added in table).
  • Chang et al., 2011: “Homotopic organization of essential language sites in right and bilateral cerebral hemispheric dominance” (reference [70])
  • Hartwingsen et al., 2014: The right posterior inferior frontal gyrus contributes to phonological word decisions in the healthy brain: Evidence from dual-site TMS” (reference [68])

Q2: Line 10 – In Abstract, moder neuro oncology – modern (typo).

R2: correction has been made.

Q3: Line 12 – In Abstract, examines literature – literatures (plural).

R3: As the Reviewer correctly state (Q15), the noun literature can be countable or uncountable. In more general, commonly used, contexts, the plural form will also be literature. However, in more specific contexts, the plural form can also be literatures e.g. in reference to various types of literatures or a collection of literatures. In our study we refer to a single body of literature e.g. that regarding the studies on TMS and neuro oncology. As such we preferred to use the word in its singular form.

Q4: Line 14 – Literature reveals that – Literatures (many literatures right?)

R4: Same as R3

Q5: Line 26 – whith the aim – with the aim. Sentences must be simplified.

R5: The sentence has been simplified and corrected: “Thus, the functional neuro-oncology aims at studying and preserve brain functions in order to ensure a better quality of life

Q6: Line 30 to 44 - Paragraphs are too long. Split the paragraphs and rewrite.

R6: Paragraphs have been divided and sentences changed. “Recent brain mapping methods have subsequently proposed the connectionist model. According to such model the large intra- and interindividual clinical variability and the different therapeutic response to similar brain damage depend on the link between cortical and subcortical areas [4]. These properties are explained by neuroplasticity i.e., the brain’s ability to reorganize in face of acquired damage or tissue removal

Q7: Line 47 to 65 (now 43-129) - Sentence connectivity is lost. No flow in reading. The writing must be in flow. Need to split the paragraphs and rewrite.

R7: We split paragraphs and rephrased sentences: “Preoperative brain mapping methods, along with cognitive evaluations, allow as-sessment of functional outcome. They allow to monitor the evolution of deficits over time and investigating the association between loss of function and plastic adaptations [9].  Among other preoperative brain mapping methods, repetitive navigated Transcranial Magnetic Stimulation (rnTMS) represents an important advance in the field of neu-ro-oncology. Indeed, it allows both mapping eloquent areas prior to surgery and guiding intraoperative direct electrical stimulation (DES) of cortico-subcortical areas [10]. The gradual introduction of TMS in neurosurgery has been the result of important scientific development. First, recent evidence suggest that functional imaging does not fully meet accuracy requirements for preoperative planning of language mapping [11]. Second, a theoretical shift from the localizationist approach to the connectivity model occurred. As such, function is now conceptually represented in extended and dynamic networks [12]. Finally, the technical improvement derived from the combination of brain stimulation techniques and neuronavigation has added more attractiveness to TMS

Q8: Line 76 to 83 (now 148-162) - Provide the introduction first which must be from standard references. Then reorganize the whole paragraph.

R8: We agree with the reviewer that a broader introduction would have improved the paragraph. Accordingly, we included a brief paragraph with standard references “The neurobiology of language has a long history that begins with the studies of the 19th century conducted by Paul Broca. Later, Wernicke’s doctoral thesis, Lichteime’s diagram and Geschwind reconceptualization of language mechanism allowed for sig-nificant contributions to the era of language [13]. Broca-Wernicke-Geschwind model provided substantial innovation to the conceptualization of language and included, in addition to the classical Broca and Wernicke areas, the involvement of white matter bundles connecting these two cortical regions [14]. The 20th century represented a step forward in the study of language neural architecture thanks to the development of structural imaging technologies with computed tomography and magnetic resonance imaging. These advances made it possible to study pathology in vivo and contributed substantially to the birth of the neurobiology of language, resulting in a confluence of different disciplines. Although Broca’s studies marked the history of classical aphasiology, with the introduction of these new techniques it has been possible to demonstrate that the areas involved in language processing extend to a large part of the subcortical bundles”

Q9: Line 111 – Figure 1: Provide reference unless it is own diagram.

R9: The figure is original

Q10: Line 118 (now 436) – the most widely used “method”? Hanging sentence there…

R10: We added the word “method”.

Q11:Line 131 – Delete the reference [1] as it is also cited in line 133.

R11: The reference has been removed

Q12: Line 143 - Figure 2: Provide reference unless it is own diagram.

R12: The figure is original

Q13: Line 163 to 187 (now 482-505) – The sentences must be continuous for easier understanding. Rewrite and organize.

R13: The sentences have been changed and paragraph reformulated “Direct electrical stimulation (DES) is performed during awake craniotomy, i.e., an anesthesia technique that allows patients to regain consciousness during surgery: it consists of electrode stimulation of exposed fiber bundles and cortical portions while performing a cognitive task during surgical intervention [31]. The “2 out of 3” rule is generally used to determine whether an area is eloquent or non-eloquent for language (or any other function) and it implies that a cortical area is stimulated three times during the execution of a task. If a performance error occurs at least twice, during DES stimulation, then that area is considered eloquent for the function tested [32]. In order to better define eloquent and non- eloquent areas, a recent study has proposed a three-level system based on cortical, subcortical and clinical features of language eloquence to allow more accurate comparison of language tumor [for more details, see 34]. 

Many studies have confirmed the accuracy of DES compared with other noninvasive methods. In particular, comparisons between fMRI and DES have yielded conflicting results, partly due to the methodological differences of these two methods. fMRI is based on statistical analysis of regional changes in oxygenated blood which, in presence of intracerebral lesions, is impaired: this hampers the accuracy of fMRI [11]. In light of these findings, DES and also rnTMS, already replace fMRI in some institutions [34]. However, DES, besides being invasive, has additional disadvantages, such as patient noncom-pliance. In these cases, for a successful intraoperative mapping, managing patient ex-pectations is vital [35]. Despite its limitations and invasiveness, to date, DES remains the gold standard since it allows defining the functional boundaries of resection in glioma surgery [36]. However, for those neuro oncological patients who are not candidates for awake craniotomy, in order to guide surgical resection, it’s important to identify a standardized and safe preoperative mapping protocol [37], especially for those with language tumors

Q14: Line 282 to 307 – The sentences must be written with flow in line with previous text. Separate into 2 or 3 paragraphs.

R14: Sentences changed and divided into 3 paragraph: “Modern neuroscience has increasingly enabled the identification of areas involved in language as a previous study which suggested that the posterior middle frontal gyrus (MFG) is an important center of cortical integration for dorsal and ventral flows of lan-guage [56]. More importantly, they demonstrated that MFG is a cluster of positive re-sponses (language errors elicited by DES) in the MFG to both rnTMS and DES stimulation in tumor patients [56].

The previous studies mentioned refer to cortical tumor. In the past decades, sub-cortical representation of language has received increasing attention.  In fact, cases with damage to subcortical traits show more severe language loss than those with a cortical damage [57]. However, rnTMS is not suitable for investigating deep subcortical structures because of its physical limitations [42].

Recent innovations have enabled subcortical mapping through the combination of rnTMS and DTI-FT. For this type of tracking, information in the regions of interest (ROIs) is needed to identify start and/or end points at the cortical level that are connected by the subcortical tracts. Specifically, rnTMS stimulation of cortical areas allows positive/eloquent areas to be identified and defined as ROIs, in order to visualize subcortical tracts with subsequent tractography [10]. The combination of rnTMS with DTI- FT has been suc-cessfully refined in recent years leading to remarkable results also from a clinical per-spective [58]. Indeed, neuro-oncology patients undergoing rnTMS-based DTI-FT showed fewer deficits at discharge, compared to the control group [59].

A study using nrTMS and DTI-FT raised the question of whether it’s possible to achieve a better visualization of fiber tracts by adding the action naming task [60]. The authors showed that the action naming task with rnTMS mapping and subsequent tracking led to a better visualization of the language subcortical network in healthy volunteers. Moreover, mapping and tractography with object and action naming is particularly useful for studying brain reorganization induced by tumor growth [60]. Another study evaluated the reliability of rnTMS in mapping functional architecture of AF in patients with tumors in the perisylvian regions. Authors showed that language errors occurred when stimulating mainly the frontal and parietal regions and corre-sponded in all cases with at least one anatomical termination of the AF. These findings suggest that the rnTMS-based DTI-FT protocol can be useful to investigate the subcortical architecture of language-related areas in cancer patients [61].”

Q15: Line 360 (now 905) – Literature reveals that - The noun literature can be countable or uncountable. In more general, commonly used, contexts, the plural form will also be literature. However, in more specific contexts, the plural form can also be literatures e.g. in reference to various types of literatures or a collection of literatures.

R15: Same as R3

Q16: Line 360 to 388 - Split the paragraphs for better understanding.

R16: We split the paragraph in 3 sections

Q17: Line 401 to 405 (now 950-954) – Rewrite the sentences.

R17: Sentences have been rewritten: “Taken together, these results suggest that in the near future language tumor re-section could be driven solely by rnTMS language mapping. This will avoid high costs, invasiveness and ineligibility of patients to intraoperative DES. This requires further studies and a consensus meeting in neuro oncology community about the best protocol to map eloquent language areas compromised by tumors.”

Q18: Line 407 to 416 (now 955-974) – Reorganize the sentences to improve the clarity.

R18: We reorganized the sentences and added additional limitations about search strategy: “Considering that this is not a systematic review, it has inherent limitations.

Although every effort has been made and selection criteria have been defined, it’s possible that some studies may not have been included in the present review, for ex-ample, non-English articles and studies that have been published in non-article form (e.g. conference papers). Our aim was to provide a narrative review focused on navigated rnTMS language mapping applied only to patients with tumor in language-related areas. Our purpose, together with the selection criteria, restricted the eligibility of studies. Nevertheless, we thought it would be useful to contribute to a general overview of such a specific issue of rnTMS developments over the past decade. Future studies could use this review as reference point about the current language protocols to investigate language subcomponents.

Moreover, the heterogeneity of findings among cited studies could be related to the different methodologies employed. Indeed, studies used different sample selection, re-cruitment criteria and outcome measures. In particular, the age range vary, the heter-ogeneity of pre, intra and post language assessments results may depend on concomitant adjuvant therapy, tumor grade, extent of infiltration and language reorganization. Such variables should be carefully taken into account in order to accurately estimate the postoperative sparing of language skills. Resolving these issues requires consensus in the awake surgery/neuro-oncology community, for which some attempts have been made in Europe [79].”

Q19: Line 418 to 424 – Write in simple sentences.

R19: Sentences has been simplified: “The shift from the localizationist model to dynamic and interconnected network models has allowed over time a multidisciplinary approach of neuro-oncological patients. Future studies can deepen the complexity of brain network by adding tasks that evaluate also the linguistic subcomponents.”

Q20: Table 1 and Table 2 – The literatures were arranged from latest year 2022 to year 2013. However, the fourth row is [34] from year 2015. Any reason for that?

R20: [34], now [33], corresponds to this reference:  Ille, S.; Schroeder, A.; Albers, L.; Kelm, A.; Droese, D.; Meyer, B.; Krieg, S.M. Non-Invasive Mapping for Effective Preoperative Guidance to Approach Highly Language-Eloquent Gliomas-A Large Scale Comparative Cohort Study Using a New Classification for Language Eloquence. Cancers (Basel). 2021, 13, 207, doi:10.3390/cancers13020207.

Q21: How did the authors do the search? First search in Google Scholar (custom range of 2013-2022) using all keywords gave return of 2770 results. Did the authors read all 2770 papers or just the first 3 pages?

R21: We read all the articles (title and abstract) fixed with mentioned keyword until results had a weak relationship with our search focus (usually from page 5 to 10). Then we applied our selection criteria (now expanded in section 2) which had significantly restricted the study selection. Our purpose was to provide a focused narrative review, based only on navigated TMS, only language mapping, only language task and only applied to patients with tumor in language-related areas. Also, non-English articles, articles that did not produce results and therefore were never published were excluded. Therefore, the criteria we fixed causes exclusion of several articles. In addition, this is not a systematic review so it has inherent limitations in itself. To be clearer, we refined the methodological part and add a paragraph in “limits”.

Q22:For PubMed, search was conducted using ((((TMS) AND (navigated TMS)) AND (neuro oncology)) AND (language)) AND (preoperative mapping) keywords and gave only one result. Alternatively, another search was conducted using ((((TMS) OR (navigated TMS)) OR (neuro oncology)) OR (language)) OR (preoperative mapping) keywords and gave 435,386 results. How did the authors conduct the search?

R22: In pubmed we applied different searches and then combined the results:

  • Navigated TMS and language and brain tumor (31 results)
  • Navigated transcranial magnetic stimulation and language and brain tumor (73 results)
  • Navigated transcranial magnetic stimulation and Direct Electrical Stimulation and language and brain tumor (11 results)

Q23: Overall - Write in simple sentences and check spellings with grammatical errors in the whole manuscript.

R 23: Thank you for your suggestions. We checked for spelling errors through the whole manuscript

Reviewer 2 Report

This review takes preoperative brain mapping as the key topic, and summarizes its important role in identifying the surgical site of neurosurgery and improving the clinical outcome by synthesizing the existing research results. After reading this manuscript, I have no questions about the content of this article. The ideas and logic of the article are relatively clear. Although rnTMS has been widely used in clinical practice, the article is not very innovative, but it is also worth publishing to provide reference opinions for scholars in the field.

Author Response

We thank the Reviewer for his/her appreciation of our study

Reviewer 3 Report

The manuscript is focused on transcranial magnetic stimulation in brain tumor-related language mapping. The authors thoroughly explained the procedure how they selected literature/studies for compiling this review. It is surprised to see that there are only 18 papers/studies related to this topic, excluding reviews.

It is worth to note that rnTMS application in neurosurgical procedure and effectively reducing postoperative language problems. The authors also discussed the limitations of rnTMS. The authors presented the data in detailed and in an easy understandable way. I thoroughly enjoyed reading this manuscript. The review will be of interest to the readers of not only this journal, but to the general readers as well. 

However, there are several typos that needs to be corrected such as moder, whith, patient,s  etc..

Author Response

We thank the Reviewer for his/her appreciation of pur study. We checked the entire manuscript for typos and errors

Round 2

Reviewer 1 Report

The authors have satisfactorily addressed the issues and responded to all my questions and made the necessary changes to the manuscript.